# Characteristics of Internal Ammonium Loading from Long-Term Polluted Sediments by Rural Domestic Wastewater

**DOI:** 10.3390/ijerph16234657

**Published:** 2019-11-22

**Authors:** Xiang Luo, Yungui Li, Qingsong Wu, Zifei Wei, Qingqing Li, Liang Wei, Yi Shen, Rong Wang

**Affiliations:** 1Department of Environmental Engineering, Southwest University of Science and Technology, Mianyang 621010, China; lx_luoxiang@163.com (X.L.); w1804527110@163.com (Q.W.); wezifei@163.com (Z.W.); weiliang1031@126.com (L.W.); 2Sichuan Provincial Sci-Tech Cooperation Base of Low-cost Wastewater Treatment Technology, Southwest University of Science and Technology, Mianyang 621010, China; 3Stockbridge School of Agriculture, University of Massachusetts, Amherst, MA 01003, USA; 4Department Key Laboratory of Microbial Technology for Industrial Pollution Control of Zhejiang Province, College of Environment, Zhejiang University of Technology, Hangzhou 310032, China; shenyi@zjut.edu.cn; 5School of National Defense Science and Technology, Southwest University of Science and Technology, Mianyang 621010, China; wr276432218@163.com

**Keywords:** ammonium, sediment, dredging, rural domestic wastewater, small basin

## Abstract

Given long-term decentralized and centralized rural domestic wastewater (RDW) discharge, nitrogen is continuously depositing in sediments. RDW discharge is assumed to be an important source of ammonium in surface water; however, the effect of long-term RDW discharge on nitrogen pollution in sediments remains unknown. Batch incubations were conducted to investigate the characteristics of internal ammonium loading from long-term polluted sediments by RDW discharge. Four sediments were demonstrated to be heavily polluted by long-term RDW discharge, with total nitrogen (TN) values of 5350, 8080, 2730, and 2000 mg·kg^−1^, respectively. The internal ammonium release from sediment was a slow and long process, and the risk of ammonium release from sediment during the dry season was significantly greater than that during the wet season. Though all selected sediments were heavily polluted by long-term RDW discharge, the relative contribution of internal ammonium loading from sediments was generally lower than that of external pollution. Hence, dredging is not suggested for RDW-polluted sediments except in response to an emergency. The excessive ammonium in the selected catchment was mainly from untreated and centralized black water in RDW. Centralized black waters in rural communities are highlighted to be separately treated or reused to maintain ammonium content at a safe level.

## 1. Introduction

The nitrogen cycle is among the most significant of earth’s biogeochemical cycles. It has been radically disrupted by the excessive increase in anthropogenic nitrogen discharge, which totals approximately 2.7 times the estimated safe nitrogen discharge threshold in China [1]. Along with tightly controlled point-source pollution and urban nonpoint source (NPS) pollution in China, rural NPS pollution is increasing, and rural domestic wastewater (RDW) discharge is responsible for 8% of the nitrogen discharge [1,2,3]. Pervasive RDW is the key sources of surface water pollution in rural areas [4,5], where a large amount of highly concentrated nitrogen is directly discharged into nearby surface water bodies [6,7]. Most RDW is decentralized discharge because of the lack of wastewater collection systems. With the rapid development of the rural economy, rural households are more centralized within rural communities, where wastewater collection systems are generally designed and built as encouraged by the policy of local governments, resulting in the relatively centralized RDW discharge in China.

Given long-term decentralized and centralized RDW discharge, nitrogen is continuously accumulating in the sediments of small basins in rural areas, such as low-lying lands, ponds, and small water catchments. However, the effect of long-term decentralized and centralized RDW discharge on nitrogen pollution in the sediments of small basins remains unknown. Sediments can serve as both a sink and source of nitrogen [8,9]. Notably, sediment release (also as NPS pollution) is another important source for aqueous nitrogen pollution [10,11,12,13,14,15,16]. Much attention has been paid to water bodies in which sediments have been polluted by integrated sources, including both point-source and NPS pollutions [1,11,17,18,19,20]. For example, Dianchi Lake [8], Chaohu Lake [21], and Erhai lake in China [22], and the Mississippi River in the United States [23]. However, the internal loading of nitrogen from sediments that have been chronically contaminated by decentralized and centralized RDW discharge (NPS pollution) in the small basins of rural areas (ponds or wastewater catchments) remains unknown. Furthermore, the role of internal loading from sediments and RDW discharge in the aqueous nitrogen balance in rural small basins has not been evaluated.

In the present study, two ponds and two small water catchments that have been subject to long-term pollution by RDW discharge (decentralized and centralized) were selected to assess the effect of this discharge on nitrogen pollution in sediments. Ammonium was chosen as the representative reactive nitrogen pollutant. Batch incubation in ultrapure and sample water was conducted to first investigate characteristics of internal ammonium loading from long-term polluted sediments by RDW [24,25].

## 2. Materials and Methods

### 2.1. Sample Collection

Four sediments (S1–S4) and related sample waters (SW1–SW4) were collected from four small basins (B1–B4), all of which were polluted by RDW. The site information of B1–B4 is listed in Table 1. RDW mainly includes black water (sewage from toilets) and gray water (other domestic wastewater except that from toilets, including from cooking, dishwashing, and baths) [26]. S1 and SW1 were sampled at a small fish pond (B1) near farmland which has been polluted by decentralized untreated RDW (gray water) discharge for more than 30 years. S2 and SW2 were sampled from a landscape pond (B2) with treated centralized RDW (both black and gray water) input over the most recent 20 years. S3/SW3 and S4/SW4 were sampled from two small water catchments (B3 and B4) around small scale rural communities. B3 has been polluted by centralized RDW discharged from a 12-household community for more than 20 years but only by gray water. B4 has been polluted by centralized RDW containing both gray and black water discharged from an eight-household community for nearly 10 years.

A multiple mixture sampling method was used to collect sample water (SW1–SW4) from the surface layer of B1–B4 for determination of ammonium and incubation experiments. Each sediment sample (S1–S4) was collected from five random spots along the shore of the selected basin with a sampling grab at a depth of 0–10 cm and then mixed homogenously. Larger objects such as gravel, leaves, and organic debris were removed from fresh sediments. Some wet and fresh sediments were used for the incubation experiments. Other sediments were freeze dried after being transported to the laboratory in sealed plastic bags placed in iceboxes. After freeze drying, sediment was milled and sieved through a 100 mm screen. The samples were sealed and stored for characterization.

### 2.2. Sample Characterization

Ammonium concentrations in four sample waters (SW1–SW4) were determined after sampling in these small basins. The pH values of sediments (S1–S4) were measured using a pH meter (PHS-2C Precision Acidity Meter, Shanghai Jingke Leici, Shanghai, China) by mixing the sediments with ultrapure water at a sediment-to-water ratio of 1:2.5. The sediment water content was measured using the drying method. The total organic carbon (TOC) contents in S1–S4 were measured using a total organic carbon analyzer (Elementar Analysensyteme GmbH, Hanau, Germany). The sediment nitrogen content (TN) was determined using a CHNS elemental analyzer (Elementar Analysensyteme GmbH, Hanau, Germany). The exchangeable nitrogen (TN_exc_) contents in the sediments, which includes extractable ammonium (NH_3_-N_exc_), exactable nitrate (NO_3_-N_exc_), and nitrite (NO_2_-N_exc_) were extracted by potassium sulfate (K_2_SO_4_) and measured. Ammonium (HJ535-2009), nitrite (GB7493-87), and nitrate (HJ/T346-2007) were determined according to the Chinese Standard using an ultraviolet spectrophotometer (UV-1600 Shanghai Mapada Instruments Co., Ltd., Shanghai, China). The detection limit was 0.025 mg·L^−1^ for ammonium, 0.003 mg·L^−1^ for nitrite, and 0.08 mg·L^−1^ for nitrate, respectively.

### 2.3. Sediments Incubation

The ammonium release characteristics from sediments were investigated by simulated laboratory batch incubation in ultrapure water. The role of internal loading from long-term RDW-polluted sediments and the RDW discharge on ammonium balance was illustrated by batch incubation in sample water. The experimental design is shown in Table 2. The incubation experiment was performed in a 15.6 L polyethylene barrel. During all batch tests, the initial sample water volume was maintained at 10.0 L and the weight of the wet sediment was adjusted according to the designed solid-to-liquid ratio. The tested sediment was spread at the bottom of a barrel and the liquid phase was gradually introduced from the top of the sediment using a siphoning tube. The sediment was evened out allowing particulate settling prior to the experiments. During the experiment, the outside of the barrel was wrapped with black bags and only daylight was allowed to contact the tested system from the top to mimic the sediment environment. Incubation was controlled at a natural temperature and under open-air conditions.

The study of different sediment-to-water ratios on the influence of ammonium release from sediment was conducted using the S1 sediment. Sediment-to-water ratios of 1:20, 1:10, 1:5, and 1:1 were used. In the studies of the ammonium evolution in the sample water of sediments from different waterbodies, the sediment-to-water ratio was maintained at a 1:20 ratio, designed to mimic the hydrologic condition. In the study of each sediment, four barrels were used in parallel as follows: (1) sediment and 10.0 L of sample water from a studied waterbody; (2) sediment with 10.0 L of pure water (Milli-Q) with the same sediment-to-water ratio as that in (1); (3) 10.0 L of ultrapure water alone to monitor the cleanliness of the atmosphere; and (4) 10.0 L of sample water alone to monitor the natural evolution of ammonium in the sample water without sediment. At a given incubation time, an aliquot of 50.0 mL of supernatant sample water was taken from each barrel and the water samples were filtered through a 0.45 µm filter. Their ammonium concentrations were determined and termed $C_{\mathrm{ssw},\;t}$, $C_{\mathrm{spw},\;t}$, $C_{\mathrm{pw},\;t},$ and $C_{\mathrm{sw},\;t}$, respectively. A duplicate sample was periodically analyzed for ammonium.

## 3. Results and Discussion

### 3.1. Ammonium Pollution of Sample Waters Caused by RDW

The classification of aqueous ammonium in small basins (SW1–SW4) according to the Chinese environmental quality standards for surface water is shown in Figure 1. The ammonium concentrations were relatively low in SW1–SW3 at 0.49 mg·L^−1^, 0.72 mg·L^−1^, and 0.71 mg·L^−1^, respectively. The water quality of SW1, SW2, and SW3 was determined to be Class Ⅱ, Class Ⅲ, and Class Ⅲ, respectively; where Class Ⅱ ≤ 0.5 mg·L^−1^ and Class Ⅲ ≤ 1.0 mg·L^−1^. The ammonium levels were 42.0 mg·L^−1^ in SW4, which significantly breached the Class Ⅴ surface water-quality standard of ≤ 2.0 mg·L^−1^.

### 3.2. Characterizations of RDW-Polluted Sediments

Selected physicochemical properties of the four tested sediments (S1–S4) are listed in Table 3. pH values of the four surface sediments were slightly alkaline (ranging from 7.60 to 9.07). The total organic carbon (TOC) contents of S1–S4 were much different. Compared to the TOC contents of the two ponds (S1 = 24.9 g·kg^−1^ and S2 = 55.6 g·kg^−1^), the TOC contents of the two catchments (S3 = 14.5 g·kg^−1^ and S4 = 14.1 g·kg^−1^) were much lower. Total nitrogen content (TN) is a useful index for assessing sediment nitrogen pollution, which is defined as heavily polluted if TN is greater than 1500 mg·kg^−1^ [27]. The TN of S1–S4 was 5350, 8080, 2730, and 2000 mg·kg^−1^, respectively, all indicating heavy pollution and necessary dredging. Notably, TN was positively linearly related to TOC (*R*^2^ = 0.87), suggesting that the nitrogen and carbon loadings in the sediment were simultaneous processes of natural biomass input and external water pollution (Appendix A).

The risk of ammonium release from sediment mainly results from exchangeable nitrogen. Considered as releasable nitrogen [28], the exchangeable nitrogen (TN_exc_) contents ranged from 90.1 to 243 mg·kg^−1^, accounting for 3.0% to 4.7% of the TN in the sediments. The proportions of the exchangeable ammoniacal nitrogen (NH_3_-N_exc_) to TN_exc_ ranged from 48.4% to 88.7%, which were much higher values than those of the exchangeable nitrite and nitrate nitrogen. NH_3_-N_exc_ greatly varied in the order of S1 (169 mg·kg^−1^) > S2 (158 mg·kg^−1^) > S3 (71.8 mg·kg^−1^) > S4 (43.6 mg·kg^−1^), consistent with the order of TN (Appendix A).

### 3.3. Ammonium Release Characteristics of RDW-Polluted Sediments

The fishpond sediment (S1) was selected to investigate the effect of the sediment-to-water ratio on internal ammonium release from sediment incubation in ultrapure water (Figure 2). $C_{\mathrm{spw},\;t}$ represents the ammonium concentration with time in the ultrapure water–sediment incubation system (mg·L^−1^), while $C_{\mathrm{pw},\;t}$ (mg·L^−1^) is the ammonium level with time in ultrapure water without sediments, which was near zero during the trial period. Therefore, the intrinsic internal ammonium release ($C_{\mathrm{spw},\;t}^{\prime }{\;=\;C}_{\mathrm{spw},\;t}\;–\;C_{\mathrm{pw},\;t}$) was nearly equal to $C_{\mathrm{spw},\;t}$. The kinetics of $C_{\mathrm{spw},\;t}$ were used to illustrate the ammonium release potential from four RDW-polluted sediments and the effect of the sediment-to-water ratio during the incubations. Ammonium can be released from the sediment by a number of processes such as diffusion, desorption, and transformation.

#### 3.3.1. Effect of the Sediment-to-Water Ratio

According to Figure 2, the kinetics of $C_{\mathrm{spw},\;t}$ were divided into three stages: A rapid increase stage, a slow decrease stage, and a stable stage. During the first stage, $C_{\mathrm{spw},\;t}$ rapidly increased over 3–10 days which was mainly because of the internal release of the RDW-polluted sediments. During the second stage, $C_{\mathrm{spw},\;t}$ gradually decreased via natural reduction processes over approximately 10 days, such as nitrification during the incubation process. During the final stable stage, $C_{\mathrm{spw},\;t}$ remained at a relatively low concentration (0.23–0.36 mg·L^−1^) for the four different sediment-to-water ratios. Despite the similar concentrations during the final stable stage, the maximum $C_{\mathrm{spw},\;t}$ ($C_{\mathrm{spw},\;m}$) values were highly influenced by the sediment-to-water ratios during the incubations (Appendix A). The time to reach the maximum $C_{\mathrm{spw},\;t}$ ($C_{\mathrm{spw},\;m}$) gradually increased with an increase in the sediment-to-water ratios, ranging from 3–10 days (Figure 2).

$C_{\mathrm{spw},\;m}$ is a useful index for an internal ammonium release risk comparison. During S1 incubation, it increased with an increase in the sediment-to-water ratio (i.e., 1.42, 1.99, 2.07, and 3.36 mg·L^−1^, at sediment-to-water ratios of 1:20, 1:10, 1:5, and 1:1, respectively), all of which breached the Class Ⅲ surface water-quality standard in China (Figure 2). This result suggests that the risk of ammonium release from sediment during the dry season (November to April in Mianyang) is significantly greater than that during the wet season (May to October in Mianyang). The ammonium concentration during the dry season was also found to be greater than that during the wet season in the same hydrological years [29]. Therefore, we should pay much more attention to the ammonium concentration during the dry season which should be constantly monitored and strictly controlled.

#### 3.3.2. Variations in the Ammonium Release of Four Heavily Polluted Sediments

The kinetics of ammonium release from the S1−S4 incubation in ultrapure water is shown in Figure 3. The ammonium release kinetics of S1−S4 in the ultrapure water–sediment incubation system was similar to 3.3.1, including three stages. Despite the similar balance concentration in the final stable stage, the maximum of $C_{\mathrm{spw},\;t}$ ($C_{\mathrm{spw},\;m}$) was much different among the four RDW-polluted sediments. The time to reach the maximum of $C_{\mathrm{spw},\;t}$ ($C_{\mathrm{spw},\;m}$) was similar for S1−S4 (Figure 3). Though the four chosen sediments were all heavily polluted (TN > 1500 mg·kg^−1^), the $C_{\mathrm{spw},\;m}$ values of S1–S4 were relatively low when the sediment-to-water ratio was 1:20, in the order of S1 (1.42 mg·L^−1^, Class Ⅳ) > S3 (0.69 mg·L^−1^, Class Ⅲ) > S4 (0.55 mg·L^−1^, Class Ⅲ) > S2 (0.50 mg·L^−1^, Class Ⅱ). Comparing the $C_{\mathrm{spw},\;m}$ of S1–S4 to the ammonium concentration in SW1–SW4, we found that the $C_{\mathrm{spw},\;m}$ of S1 was much higher than that of SW1 (0.49 mg·L^−1^), and the $C_{\mathrm{spw},\;m}$ of S2–S3 was similar to the ammonium in SW2–SW3. The $C_{\mathrm{spw},\;m}$ value of S4 was dramatically less than the ammonium in SW4. This demonstrated that the important ammonium in SW1–SW3 was from internal loading, but the ammonium in SW4 was mainly attributed to the centralized black water discharge. The lower ammonium concentration in SW1 (0.49 mg·L^−1^) was because the real sediment-to-water ratio in pond B1 was less than 1:20. It has been suggested that black waters in centralized RDW be separately collected for further treatment or resource cycle [1].

#### 3.3.3. The Difference of Ammonium Release between Theoretical Calculation and Testing

Notably, the order of $C_{\mathrm{spw},\;m}$ (S1 > S3 > S4 > S2) was much different from that of TN and NH_3_-N_exc_ (i.e., S2 > S1 > S3 > S4). This suggests that it is difficult to use TN or NH_3_-N_exc_ to directly evaluate the ammonium release risk from polluted sediments. Assuming that all ammonium in a sediment is released into 10.0 L of ultrapure water, its concentration in such a system ($C_{\mathrm{spw}}^{c}$) can be expressed by Equation (1) as follows, and as shown in Appendix A.
(1)$$C_{\mathrm{spw}}^{c}\;=\;\frac{w\;\times \;(1-M\%)\;\times \;m}{V}$$
where $C_{\mathrm{spw}}^{c}$ (mg·L^−1^) is the theoretical ammonium concentration in an ultrapure water–sediment incubation system, w (mg·kg^−1^) is the exchange ammonium nitrogen content in a dry sediment, *M* (%) is the water content of the tested sediments, *m* (kg) is the mass of the experimental sample, and *V* (L) is the volume of the water. The $C_{\mathrm{spw}}^{c}$ values were found to be much greater than those of $C_{\mathrm{spw},\;t}$, and this gap was more distinctive when the incubation was at a higher sediment-to-water ratio. For example, $C_{\mathrm{spw}}^{c}$ of S1 reached 56.3 mg·L^−1^ when the sediment-to-water ratio was 1:1, which was 17-fold higher than the $C_{\mathrm{spw},\;m}$ of S1. This indicated that the internal ammonium release from sediment was a slow and long process and that the risk of ammonium release can be overestimated by the theoretical calculation using NH_3_-N_exc_, particularly during the dry season.

### 3.4. Characteristics of the RDW-Polluted Sediments Incubated in Sample Water

#### 3.4.1. The Source and Sink Effect of Sediment on Ammonium Balance

The kinetics of ammonium with S1 incubation at different sediment-to-water ratios in sample water are shown in Figure 4 and those of the S1–S4 incubation in sample water are shown in Figure 5. Sediments can act as source or sink in the aquatic ecosystem cycle and are important in regulating the ammonium balance [30]. The intrinsic contributions from sediments to aqueous ammonium in the sample water–sediment incubation system ($C_{\mathrm{ssw},\;t}^{\prime }$) were calculated using Equation (2) and are shown in Appendix A:(2)$$C_{\mathrm{ssw},\;t}^{\prime }{\;=\;C}_{\mathrm{ssw},\;t}\;{-\;C}_{\mathrm{sw},\;t}$$
where $C_{\mathrm{ssw},\;t}$ (mg·L^−1^) is the ammonium concentration with time in the sample water–sediment incubation. Sediment was a source of ammonium when $C_{\mathrm{ssw},\;t}^{\prime }$ > 0, while it was a sink with the dominant ammonium sorption onto sediment when $C_{\mathrm{ssw},\;t}^{\prime }$ < 0. The calculated intrinsic contributions from sediments ($C_{\mathrm{ssw},\;t}^{\prime }$) in the sample water were generally slightly lower than those ($C_{\mathrm{spw},\;t}^{\prime }$) in ultrapure water (except as shown in Appendix A), mainly given the stronger natural ammonium reduction in sample water with abundant nitrifying bacteria.

Similar to the kinetics of $C_{\mathrm{spw},\;t}$ in the ultrapure water–sediments incubations, the kinetics of $C_{\mathrm{ssw},\;t}$ also included three stages (i.e., first a rapid increase, then a slow decrease, and finally a stable stage at a relatively low concentration, except for S4 as shown in Figure 5D). These three stages were reached within approximately 13–20 days. The kinetics of $C_{\mathrm{ssw},\;t}$ in the S4 incubation with sample water was markedly different from that of the other three RDW-polluted sediments (i.e., $C_{\mathrm{ssw},\;t}$ directly decreased to 1.80 mg·L^−1^ within 16 days). The distinctive kinetic of $C_{\mathrm{ssw},\;t}$ in the S4 incubation was mainly attributed to the remarkably high initial aqueous ammonium concentration in the SW4 (42.0 mg·L^−1^). The sharp decline of $C_{\mathrm{ssw},\;t}$ in the S4 incubation was quite similar to that of $C_{\mathrm{sw},\;t}$ in the SW4 incubation, suggesting that the decline in $C_{\mathrm{ssw},\;t}$ was actually the result of the natural ammonium reduction in SW4. However, the natural reduction of ammonium in the surface water contributed by the highly concentrated RDW discharge required a long time, which was among the major reasons contributing to water eutrophication and black odorous river [8,31,32]. Generally, the $C_{\mathrm{ssw},\;t}$ value was higher than that of the $C_{\mathrm{sw},\;t}$, indicating the source effect of sediments on ammonium in the surface water. Taking the most polluted S1 as an example, the maximum $C_{\mathrm{spw},\;t}^{\prime }$ reached 1.12, 1.57, 2.58, and 3.09 mg·L^−1^ with sediment-to-water ratios of 1:20, 1:10, 1:5, and 1:1, respectively. And all of the maximum $C_{\mathrm{spw},\;t}^{\prime }$ breached the Class Ⅲ surface water-quality standard in China (Appendix A).

However, the source and sink effect of sediment on ammonium was constantly changing during the incubation process and was obviously influenced by the sediment-to-water ratio and the initial SW (aqueous ammonium in small basins) concentration. (1) When the sediment-to-water ratio was 1:1, S1 served only as a source of ammonium in the entire incubation but demonstrated a “source → sink → source” effect during the incubation at sediment-to-water ratios of 1:20, 1:10, and 1:5, respectively. (2) When the initial concentration of the discharged RDW was high, S4 was a sink for pollutants in the RDW during the first five days (i.e., $C_{\mathrm{spw},\;t}$ was slightly lower than $C_{\mathrm{pw},\;t}$). Notably, the contribution of the sink effect from sediment on surface water quality can be negligible when the ammonium concentration in the RDW discharge is extremely high. (3) The constantly changing effects of sediments on regulating the ammonium balance were caused by variations in natural temperature and ammonium reduction in sample water. For example, when the aqueous ammonium gradually decreased via natural reduction processes, the exchangeable ammonium was considered to be desorbed from sediments to the sample water resulting in a “sink → source” effect.

#### 3.4.2. Relative contribution of internal loading and external discharge

The relative contribution of internal loading from sediment (*R*_s_) and external pollution (*R*_sw_, sample water) to the aqueous ammonium distribution were further calculated using Equations (3) and (4), as shown in Figure 6 and Figure 7:(3)$$R_{\mathrm{sw}}\%=\frac{C_{\mathrm{sw}}}{C_{\mathrm{ssw}}}\times 100\;\%,$$
(4)$$R_{s}\%=\frac{C_{\mathrm{ssw}}^{\prime }}{C_{\mathrm{ssw}}}\times 100\;\%,$$
where $R_{\mathrm{sw}}$(%) is the relative contribution of sample water (external pollution) to ammonium and $R_{s}$(%) is the relative apparent contribution of sediment to ammonium in the sample water–sediment system. As shown in Figure 6, the relative contribution to aqueous ammonium was mainly from the sample water during the early stage, then mostly from the internal release of S1, and then by the sample water during the final stage. $R_{s}$ from S1 increased with an increase in the sediment-to-water ratio and the time for sediment to control water quality was highly related to this ratio. As shown in Figure 7, *R*_s_ values from four of the heavily polluted sediments by long-term RDW discharge (S1–S4) were mostly less than those of *R*_sw_ with the incubation of sediments (despite the low ammonium concentration in SW1–SW3).

#### 3.4.3. Water Quality Management Suggestions for Rural Areas

Dredging was a common restoration method to reduce internal loading of sediment and was extensively used during recent decades, having been applied since the 1970s [11,33,34,35]. However, as previously described, either the absolute internal ammonium loading or the relative contribution from long-term and heavily polluted sediments by RDW discharge was relatively low (except during the dry season). Moreover, sediments also play an important role in regulating the pollution balance by acting as a sink for external pollution in the aquatic ecosystem cycle, such as for heavy metals and POPs (persistent organic pollutants) [30,36,37,38]. Hence, dredging is not believed to be the best choice for RDW-polluted sediments for aqueous ammonium risk control. The key point to maintain aqueous ammonium at a safe level is to manage the major source of external pollution (i.e., mainly RDW-discharge and particularly centralized black water). Centralized RDW with black water should be strictly treated before discharge into surface water bodies. The small-scale sewage treatment technology for RDW is relatively mature in China and the most important and difficult process is the related operational management [39]. Considering the significant cost of regular RDW treatment is unacceptable for developing countries, centralized rural communities are highlighted, which offer the possibility of separate collection, treatment, or reuse of black water. Dredging is not suggested except in response to an emergency.

## 4. Conclusions

The risk of ammonium release from sediment during the dry season was found to be significantly greater than that during the wet season in this study. Internal ammonium release from sediment was a slow and long process and the risk of ammonium release from sediment could be overestimated by the theoretical calculation using tested exchangeable nitrogen. Sediments can serve as source or sink, regulating the ammonium balance in an aquatic ecosystem cycle such as the “sink → source” effect of a high concentration of RDW discharge. Though all of the selected sediments were heavily polluted as a result of long-term RDW discharge, their relative contribution of internal loading was generally less than that of external pollution. Hence, dredging was not suggested for RDW-polluted sediments except in response to emergency. The excessive ammonium in the selected catchment was mainly a result of discharge of untreated and centralized black water in RDW. The centralized black waters in rural communities were highlighted for separate treatment or reuse to maintain aqueous ammonium at a safe level. We call for more studies on the nitrogen cycle contributed by the RDW discharge and proper treatment technology for centralized black waters.

## Figures and Tables

**Figure 1 ijerph-16-04657-f001:**
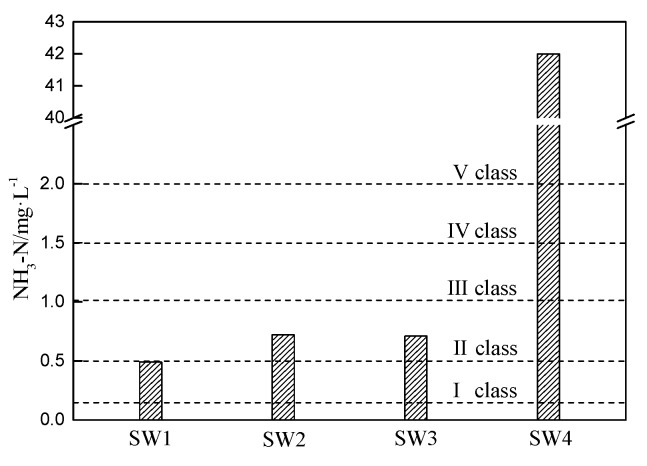
Classification of aqueous ammonium in small basin (sample water, SW1–SW4) according to the Chinese environmental quality standards for surface water (GB3838-2002).

**Figure 2 ijerph-16-04657-f002:**
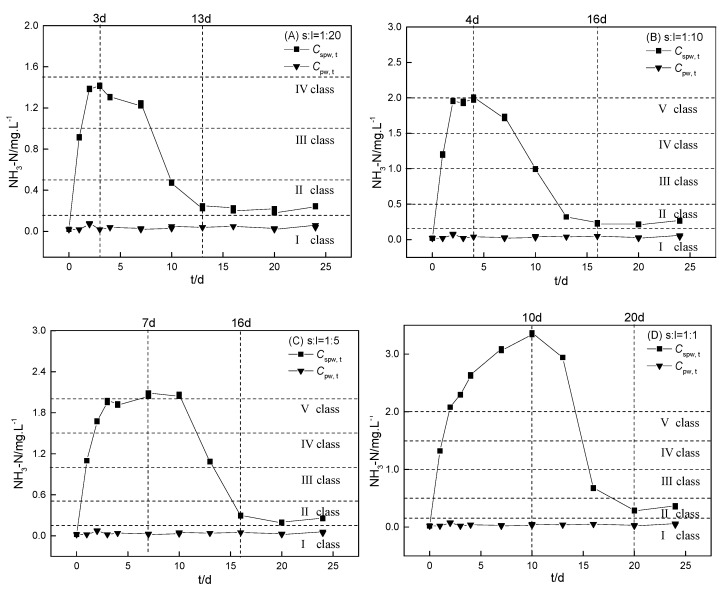
Kinetics of ammonium release from sediment (S1) incubation at different sediment-to-water ratios in ultrapure water ($C_{\mathrm{spw},\;t}$). The sediment-to-water ratios for (**A**), (**B**), (**C**), and (**D**) were 1:20, 1:10, 1:5, and 1:1, respectively. The controls were the kinetics of ammonium incubation in ultrapure water ($C_{\mathrm{pw},\;t}$) without sediments.

**Figure 3 ijerph-16-04657-f003:**
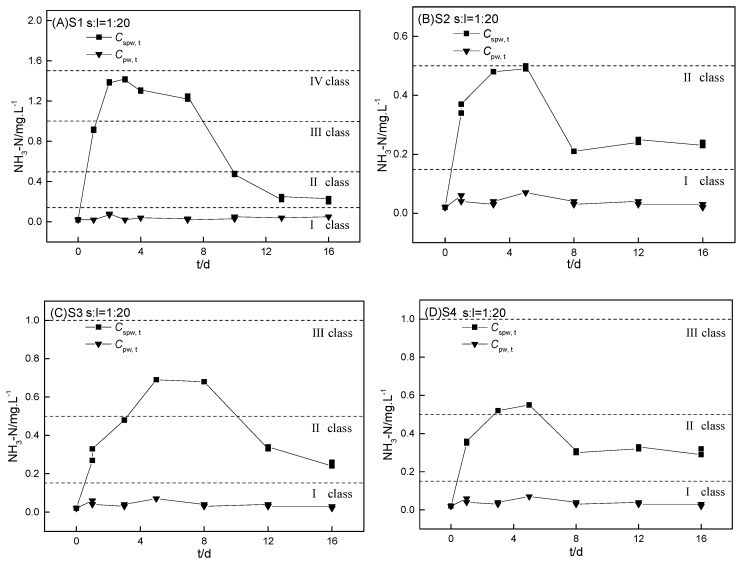
Kinetics of ammonium release from four sediments (S1–S4) incubated in ultrapure water ($C_{\mathrm{spw},\;t}$). The sediment-to-water ratio was 1:20 for the four sediments. The controls were the kinetics of ammonium incubation in ultrapure water ($C_{\mathrm{pw},\;t}$) without sediments.

**Figure 4 ijerph-16-04657-f004:**
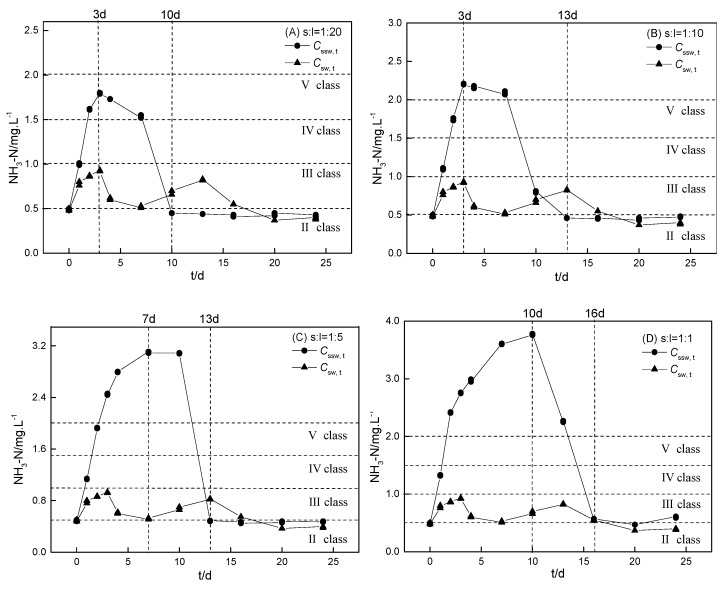
Kinetics of ammonium with sediment (S1) incubation at different sediment-to-water ratios in sample water ($C_{\mathrm{ssw},\;t}$). The sediment-to-water ratios for (**A**), (**B**), (**C**), and (**D**) were 1:20, 1:10, 1:5, and 1:1, respectively. The controls were the kinetics of ammonium incubation in the sample water ($C_{\mathrm{sw},\;t}$) without sediments.

**Figure 5 ijerph-16-04657-f005:**
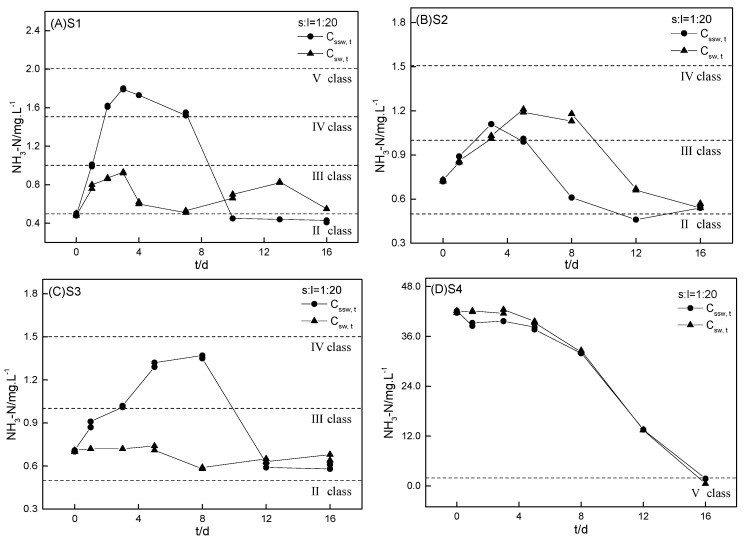
Kinetics of ammonium for four sediments (S1–S4) incubation in sample water ($C_{\mathrm{ssw},\;t}$). The sediment-to-water ratio was 1:20 for the four sediments. The controls were the kinetics of ammonium incubation in sample water ($C_{\mathrm{sw},\;t}$) without sediments.

**Figure 6 ijerph-16-04657-f006:**
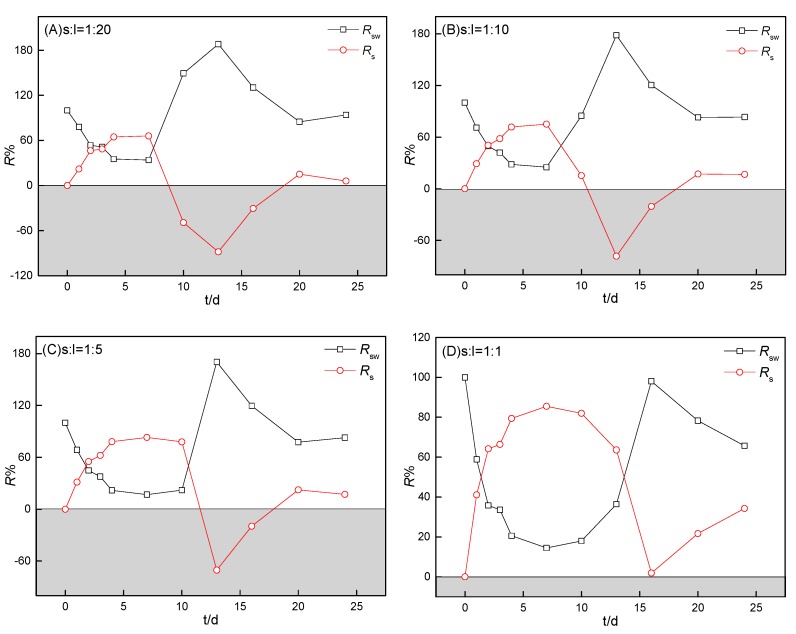
Relative contribution of internal sediment loading (S1) and external pollution (sample water) to aqueous ammonium during incubation at different sediment-to-water ratios. The sediment-to-water ratios for (**A**), (**B**), (**C**), and (**D**) were 1:20, 1:10, 1:5, and 1:1, respectively. $R_{\mathrm{sw}}$ (%) is the relative contribution of the sample water to ammonium and $R_{s}$ (%) is the relative contribution of sediment to ammonium in the sample water–sediment system.

**Figure 7 ijerph-16-04657-f007:**
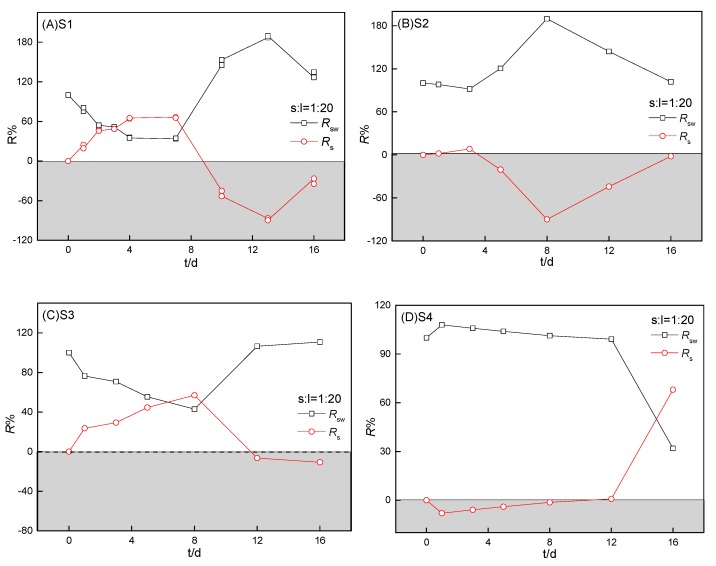
Relative contribution of internal sediment loading and external pollution (sample water) to aqueous ammonium during the four sediments’ incubation (S1–S4). The sediment-to-water ratio was 1:20 for the four sediments (S1–S4). $R_{\mathrm{sw}}$ (%) is the relative contribution of sample water to ammonium and $R_{s}$ (%) is the relative apparent contribution of sediment to ammonium in the sample water–sediment system.

**Table 1 ijerph-16-04657-t001:** Site information for the four small basins polluted by RDW (rural domestic wastewater).

Small Basin	Type	Geographic Location ^1^	Sediment Samples	Water Sample	Water Area ^2^ /(m^2^)	RDW Discharge				
						Scale	Type	Treated	History	Collection System
B1	Pond	N 31°54’56.42”, E 104°69’88.56”	S1	SW1	446	4-households	Gray water	No	30 years	No
B2	Pond	N 31°53’64.94”, E 104°70’01.99”	S2	SW2	781	>10-households	Both black water and gray water	Yes	20 years	Yes
B3	Tributary	N 31°57’79.18”, E 104°70’08.96”	S3	SW3	490	12-households	Gray water	No	20 years	No
B4	Tributary	N 31°57’86.28”, E 104°66’76.84”	S4	SW4	223	8-households	Both black water and gray water	No	10 years	Yes

Notes: ^1^ Geographic Location was determined using a Global Positioning System (GPS) in Mianyang, Sichuan Province, Southwest China. ^2^ The water area was calculated using ArcGIS software (Environmental Systems Research Institute, California, USA).

**Table 2 ijerph-16-04657-t002:** The experimental design for incubation of long-term RDW-polluted sediments.

Research Objective		Sediment ^1^	Water ^2^	Sediment-to-Water Ratio	Ammonium Concentration ^4^
Ammonium release characteristics of RDW-polluted sediments	Effect of the sediment-to-water ratio	S1	PW	1:20	$C_{\mathrm{spw},\;t}$
		S1	PW	1:10	$C_{\mathrm{spw},\;t}$
		S1	PW	1:05	$C_{\mathrm{spw},\;t}$
		S1	PW	1:01	$C_{\mathrm{spw},\;t}$
		- ^3^	PW	-	$C_{\mathrm{pw},\;t}$
	Diversity of the four heavily polluted sediments	S1	PW	1:20	$C_{\mathrm{spw},\;t}$
		S2	PW	1:20	$C_{\mathrm{spw},\;t}$
		S3	PW	1:20	$C_{\mathrm{spw},\;t}$
		S4	PW	1:20	$C_{\mathrm{spw},\;t}$
		- ^3^	PW	-	$C_{\mathrm{pw},\;t}$
The role of internal loading and external discharge on ammonium balance	Effect of the sediment-to-water ratio	S1	SW1	1:20	$C_{\mathrm{ssw},\;t}$
		S1	SW1	1:10	$C_{\mathrm{ssw},\;t}$
		S1	SW1	1:05	$C_{\mathrm{ssw},\;t}$
		S1	SW1	1:01	$C_{\mathrm{ssw},\;t}$
		- ^3^	SW1	-	$C_{\mathrm{sw},\;t}$
	Diversity of the four heavily polluted sediments	S1	SW1	1:20	$C_{\mathrm{ssw},\;t}$
		S2	SW2	1:20	$C_{\mathrm{ssw},\;t}$
		S3	SW3	1:20	$C_{\mathrm{ssw},\;t}$
		S4	SW4	1:20	$C_{\mathrm{ssw},\;t}$
		- ^3^	SW1	-	$C_{\mathrm{sw},\;t}$
		- ^3^	SW2	-	$C_{\mathrm{sw},\;t}$
		- ^3^	SW3	-	$C_{\mathrm{sw},\;t}$
		- ^3^	SW4	-	$C_{\mathrm{sw},\;t}$

Notes: ^1^ S1–S4 are the selected long-term RDW-polluted sediment samples from small basins (B1–B4); ^2^ PW is the ultrapure water (Milli-Q), and SW1–SW4 represent the selected sample waters from B1-B4; ^3^ this is the natural incubation of water without sediment. ^4^
$C_{\mathrm{spw},\;t}$ is the ammonium release from sediment incubation in ultrapure water; $C_{\mathrm{pw},\;t}$ is the ammonium incubation in ultrapure water without sediments; $C_{\mathrm{ssw},\;t}$ is the ammonium with sediment incubation in sample water; and $C_{\mathrm{sw},\;t}$ is the ammonium incubation in the sample water without sediments.

**Table 3 ijerph-16-04657-t003:** Physicochemical properties of the four tested sediments.

Sediment	pH	Water Content ^1^ /%	TOC ^2^ /(g·kg^−1^)	DOC ^3^ /(mg·kg^−1^)	TN ^4^ /(mg·kg^−1^)	TN_exc_ ^5^ /(mg·kg^-1^)	NH_3_-N_exc_ ^6^ /(mg·kg^−1^)
S1	9.07	66.6	24.9	8.3	5350	189	169
S2	8.23	85.1	55.6	7.9	8080	243	158
S3	8.37	38.3	14.5	7.5	2730	130	71.8
S4	7.60	37.3	14.1	7.3	2000	90.1	43.6

Notes: ^1^ The water contents of sediments were measured using the drying method. ^2^ TOC is the total organic carbon content, ^3^ DOC is the dissolved organic carbon content, ^4^ TN is the total nitrogen content, ^5^ TN_exc_ is the content of exchangeable nitrogen, and ^6^ NH_3_-N_exc_ is the exchange ammonium nitrogen content.

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
