# Peer review of "Characteristics of Internal Ammonium Loading from Long-Term Polluted Sediments by Rural Domestic Wastewater"

_ijerph, 2019, doi:10.3390/ijerph16234657_

Round 1
Reviewer 1 Report
The manuscript „Characteristics of Internal Loading from Long-term Polluted Sediments by Rural Domestic Wastewaters“ describes the distribution of nitrogen in sediments polluted with rural domestic wastewaters. It is nicely written and contains valuable information. However, there are some issues that should be resolved prior to its publication.
Firstly, authors need to put some extra effort into better structuring the text, separating the sections into subsections, for easier readability and comprehension.
Title should be revised. Topic is nitrogen, which is not visible from the title.
English should be improved.
Pg 2, Ln 39: „Larger sediment“ Please, rephrase.
Authors should revise the part about sediment sampling. It is unclear how the samples were taken.
Pg 3, Ln 5: how was ammonium measured?
Pg 3, Ln 27 – 38: please, revise. It is unclear how the experiment was performed. At what sediment:water ratios? How many replicates? Why was 1:20 ratio used to present kinetics of ammonium in S1-S4? This should also be explained.
Pg 4, Ln 3: SW1-SW3 which were 0.49, 0.72 and 0.71 mg·L-1 … after each number there should be a unit.
Table 3: numbers should be written with three significant figures (e.g., 0.003; 2.34; 45.6 or 302)
Pg 5, Ln 9: a verb is missing
Sections 3.3. and 3.4. should be divided into two or three subsections. It is really hard „to follow the story“ this way.
Figures position should be changed, in order to follow better the text of the manuscript.
The verb tenses in the conclusion should be checked and corrected where necessary. Specifically, it is not clear what is a generally accepted fact and what is the conclusion of the study. The same should be applied throughout the text where needed.
Also, conclusive remarks should reflect the order of discussion in the text. In this form, they seem a little scattered, which makes it difficult to see the main conclusion of the paper.
Author Response
Responses to the Reviewers Comments
#To Reviewer 1:
We sincerely appreciate for your patient reading and constructive comments on this MS. According to your valuable comments and suggestions, the MS was revised very carefully to improve the quality and readability. Please see a detailed, point-by-point response:
Firstly, authors need to put some extra effort into better structuring the text, separating the sections into subsections, for easier readability and comprehension.
Response 1: Thanks very much. For easier readability and comprehension, the sections have been separated into subsections, and the related description and discussion were revised correspondingly (as seen in Manuscript ijerph-625527-marked). Section 3.3 was divided into three subsections with 3.3.1 (Effect of the sediment-to-water ratio), 3.3.2 (Variations in the ammonium release of four heavily polluted sediments) and 3.3.3 (The difference of ammonium release between theoretical calculation and testing). Sections 3.4 was divided into three subsections with 3.4.1 (The source and sink effect of sediment on ammonium balance), 3.4.2 (Relative contribution of internal loading and external discharge) and 3.4.3 (Water quality management suggestion in rural area).
Title should be revised. Topic is nitrogen, which is not visible from the title.
Response 2: Title “Characteristics of Internal Loading from Long-term Polluted Sediments by Rural Domestic Wastewaters” was revised as “Characteristics of Internal Ammonium Loading from Long-term Polluted Sediments by Rural Domestic Wastewater”. (Pg 1, Ln 2-4)
Pg 2, Ln 39: Larger sediment “Please, rephrase.
Response 3: “Larger sediment” was changed to “Each sediment sample”. (Pg 2, Ln 39-40)
Authors should revise the part about sediment sampling. It is unclear how the samples were taken.
Response 4: For accurate description, “Sediment samples were taken with sampling grab and their surface 0-10 cm were collected. A larger sediment was collected from 5 spots along the shore, then mixed and homogenized.” was revised as “Each sediment sample (S1-S4) was collected from 5 random spots along the shore of the selected basin with a sampling grab at depth of 0-10 cm and then mixed homogenously” (Pg 2, Ln 39 - 41)
Pg 3, Ln 5: how was ammonium measured?
Response 5: The method of ammonium measured was supplemented in the text.
“Ammonium (HJ535-2009), nitrite (GB7493-87) and nitrate (HJ/T346-2007) were determined according to the Chinese Standard using a ultraviolet spectrophotometer (UV-1600 Shanghai Mapada Instruments Co., Ltd., Sha3wnghai, China), respectively.” (Pg 3, Ln 13-15)
Pg 3, Ln 27 – 38: please, revise. It is unclear how the experiment was performed. At what sediment:water ratios? How many replicates? Why was 1:20 ratio used to present kinetics of ammonium in S1-S4? This should also be explained.
Response 6: I'm sorry for this confusion because of the inaccurate description here. For better understanding, the overall experimental design for incubation of long-term RDW-polluted sediments was added in Table 2. Moreover, the related descriptions were revised as “The ammonium release characteristics from sediments were investigated by simulated laboratory batch incubation in ultrapure water. The role of internal loading from long-term RDW-polluted sediments and the RDW discharge on ammonium balance was illustrated by batch incubation in sample water. The experiment design was performed in Table 2. The incubation experiment was performed in a 15.6 L polyethylene barrel. During all batch tests, the initial sample water volume was maintained at 10.0 L and the weight of the wet sediment was adjusted according to the designed solid-to-liquid ratio. The tested sediment was spread at the bottom of a barrel and the liquid phase was gradually introduced from the top of the sediment using a siphoning tube. The sediment was evened out allowing particulate settling prior to the experiments. During the experiment, the outside of the barrel was wrapped with black bags and only daylight was allowed to contact the tested system from the top to mimic the sediment environment. The incubation was controlled at natural temperature and under open-air conditions.
The study of different sediment-to-water ratios on the influence of ammonium release from sediment was conducted using the S1 sediment. Sediment-to-water ratios of 1:20, 1:10, 1:5 and 1:1 were used. In the studies of the ammonium evolution in the sample water of sediments from different waterbodies, the sediment-to-water ratio was maintained at a 1:20 ratio which was to mimic the hydrologic condition. In the study of each sediment, 4 barrels were used in parallel as follows: (1) sediment and 10.0 L of water sample from a studied waterbody; (2) sediment with 10.0 L of pure water (Milli-Q) with the same sediment-to-water ratio as that in (1); (3) 10.0 L of ultrapure water alone to monitor the cleanliness of the atmosphere; and (4) 10.0 L of sample water alone to monitor the natural evolution of ammonium in the sample water without sediment. At a given incubation time, an aliquot of 50.0 mL of supernatant water sample was taken from each barrel and the water samples were filtered through 0.45 µm filter. Their ammonium concentrations were determined and termed , , and , respectively. A duplicate sample was periodically analyzed for ammonium.” (Pg 3, Ln 19 - Pg 4, Ln 6)
Pg 4, Ln 3: SW1-SW3 which were 0.49, 0.72 and 0.71 mg·L-1 … after each number there should be a unit.
Response 7: Thanks a lot. The unit has been added after each number. (Pg 4, Ln 13)
Table 3: numbers should be written with three significant figures (e.g., 0.003; 2.34; 45.6 or 302)
Response 8: Thanks very much. Numbers in Table 3 were revised as suggested.
Pg 5, Ln 9: a verb is missing
Response 9: “Ammonium released from the sediment by any possible processes such as diffusion, desorption, transformation.” was revised as “Ammonium can be released from the sediment by a number of processes such as diffusion, desorption, and transformation.” (Pg 6, Ln 2 - Ln 3)
Sections 3.3. and 3.4. should be divided into two or three subsections. It is really hard “to follow the story “this way.
Response 10: Thanks very much. For easier readability and comprehension, the sections have been separated into subsections. Sections 3.3 was divided into three subsections with 3.3.1 (Effect of the sediment-to-water ratio), 3.3.2 (Variations in the ammonium release of four heavily polluted sediments) and 3.3.3 (The difference of ammonium release between theoretical calculation and testing). Sections 3.4 was divided into three subsections with 3.4.1 (The source and sink effect of sediment on ammonium balance), 3.4.2 (Relative contribution of internal loading and external discharge) and 3.4.3 (Water quality management suggestion in rural area).
Figures position should be changed, in order to follow better the text of the manuscript.
Response 11: Sections 3.3 and 3.4 is divided into three subsections, respectively. So, the Figures position is consistent with the description order in the text correspondingly.
The verb tenses in the conclusion should be checked and corrected where necessary. Specifically, it is not clear what is a generally accepted fact and what is the conclusion of the study. The same should be applied throughout the text where needed. Also, conclusive remarks should reflect the order of discussion in the text. In this form, they seem a little scattered, which makes it difficult to see the main conclusion of the paper.
Response 12: Thanks very much. The verb tense in the manuscript was carefully revised as seen in Manuscript ijerph-625527-marked.
For better understanding, the conclusion was revised as “The risk of ammonium release from sediment during the dry season was found to be significantly greater than that during the wet season in this study. Internal ammonium release from sediment was a slow and long process and the risk of ammonium release from sediment could be overestimated by the theoretical calculation using tested exchangeable nitrogen. Sediments can serve as a “source” or “sink” regulating the ammonium balance in an aquatic ecosystem cycle, such as the “sink→source” effect of a high concentration of RDW discharge. Though all of the selected sediments were heavily polluted as a result of long-term RDW discharge, their relative contribution of internal loading was generally less than that of external pollution. Hence, dredging was not suggested for RDW-polluted sediments except in response to emergency. The excessive ammonium in the selected catchment was mainly a result of discharge of untreated and centralized black water in RDW. The centralized black waters in rural community were highlighted for separate treatment or reuse to maintain aqueous ammonium at a safe level. We highly call for more studies on nitrogen circle contributed by the RDW discharge and proper treatment technology for centralized black waters.”

Reviewer 2 Report
The manuscript is generally well written. However ae a few grammatical errors and typos that needs to be corrected. The introduction should emphasis further the novelty of the work as it is not very clear. The methods is scanty. If possible a diagrammatic representation f the experimental set up will enhance the clarity. Use wet season instead of plentiful season as used across the manuscript. Some precipitation data for the different season may help.Author Response
Responses to the Reviewers Comments
#To Reviewer 2:
We sincerely appreciate for your patient reading and constructive comments on this MS. According to your valuable comments and suggestions, the MS was revised very carefully to improve the quality and readability. Please see a detailed, point-by-point response:
The introduction should emphasis further the novelty of the work as it is not very clear.
Response 1: To further emphasis the novelty of the work, the final part of introduction was revised as:
“However, the internal loading of nitrogen from sediments that have been chronically contaminated by decentralized and centralized RDW discharge (NPS pollution) in the small basins of rural areas (ponds or wastewater catchments) remains unknown. Furthermore, the role of internal loading from sediments and RDW discharge in the aqueous nitrogen balance in rural small basins has not been evaluated.
In the present study, two ponds and two small water catchments that have been subject to long-term pollution by RDW discharge (decentralized and centralized) were selected to assess the effect of this discharge on nitrogen pollution in sediments. Ammonium was chosen as the representative reactive nitrogen pollutant. Batch incubation in ultrapure and sample water was conducted to firstly investigate characteristics of internal ammonium loading from long-term polluted sediments by RDW.”
The methods is scanty.
Response 2: The applied incubation experimental method was cited from the following references which have been supplemented in manuscript (Pg 2, Ln 24).
Xie, M.Z.; Chen, Q.; Dang, C.Y.; Pan, B.Y.; An, R.; Wu, Z.; Zhou, M. Study on nitrogen release from reservoir sediments and nitrogen removal by aerobic microorganism. Acta Sci Natur. 2019, 55, 168-177, doi: 10.13209/j.0479-8023.2019.018 Beutel, M.W. Inhibition of ammonia release from anoxic profundal sediments in lakes using hypolimnetic oxygenation. Ecol Eng. 2006, 28, 271-279, doi:10.1016/j.ecoleng.2006.05.009If possible a diagrammatic representation f the experimental set up will enhance the clarity.
Response 3: Thanks for your suggestion. For better understanding, the overall experimental design for incubation of long-term RDW-polluted sediments was added in Table 2. Moreover, the related descriptions were revised as “The ammonium release characteristics from sediments were investigated by simulated laboratory batch incubation in ultrapure water. The role of internal loading from long-term RDW-polluted sediments and the RDW discharge on ammonium balance was illustrated by batch incubation in sample water. The experiment design was performed in Table 2. The incubation experiment was performed in a 15.6 L polyethylene barrel. During all batch tests, the initial sample water volume was maintained at 10.0 L and the weight of the wet sediment was adjusted according to the designed solid-to-liquid ratio. The tested sediment was spread at the bottom of a barrel and the liquid phase was gradually introduced from the top of the sediment using a siphoning tube. The sediment was evened out allowing particulate settling prior to the experiments. During the experiment, the outside of the barrel was wrapped with black bags and only daylight was allowed to contact the tested system from the top to mimic the sediment environment. The incubation was controlled at natural temperature and under open-air conditions.
The study of different sediment-to-water ratios on the influence of ammonium release from sediment was conducted using the S1 sediment. Sediment-to-water ratios of 1:20, 1:10, 1:5 and 1:1 were used. In the studies of the ammonium evolution in the sample water of sediments from different waterbodies, the sediment-to-water ratio was maintained at a 1:20 ratio which was to mimic the hydrologic condition. In the study of each sediment, 4 barrels were used in parallel as follows: (1) sediment and 10.0 L of water sample from a studied waterbody; (2) sediment with 10.0 L of pure water (Milli-Q) with the same sediment-to-water ratio as that in (1); (3) 10.0 L of ultrapure water alone to monitor the cleanliness of the atmosphere; and (4) 10.0 L of sample water alone to monitor the natural evolution of ammonium in the sample water without sediment. At a given incubation time, an aliquot of 50.0 mL of supernatant water sample was taken from each barrel and the water samples were filtered through 0.45 µm filter. Their ammonium concentrations were determined and termed , , and , respectively. A duplicate sample was periodically analyzed for ammonium.” (Pg 3, Ln19 - Pg 4, Ln 6)
Use wet season instead of plentiful season as used across the manuscript.
Response 4: “plentiful season” were revised as “wet season” in manuscript. (Pg 1, Ln 28; Pg 6, Ln 20; Pg 6, Ln 21; Pg 12, Ln 9)
Some precipitation data for the different season may help.
Response 5: Thanks for your suggestion. The characteristics of the local precipitation have been supplemented as “This result suggested that the risk of ammonium release from sediment during the dry season (November to April in Mianyang) was significantly greater than that during the wet season (May to October in Mianyang).” (Pg 6, Ln 19 - 20).

Reviewer 3 Report
This is good scientific manuscript making an important contribution for knowledge advancement on the degree of ammonium / nitrogen accumulation in the small basins mainly because centralised or decentralised of wastewater management
I checked the plagiarism status, happy to report that its very low, therefore suggesting the high originality of this work.
However, the language used is of low quality therefore after making revisions, please submit the whole manuscript to a language editor for refinements and editorials
The rationale and justification for this study is not clearly specified, is this a pilot study because you collected only four samples to represent the whole basin or catchments. I suggest that you give a justification of this limited sampling. Maybe its a pilot study and will be followed by a bigger study, however that should be clearly identified and justified upfront.
The presentation of the results needs improvement. The concentration of pollutants expressed in the results must also be interpreted in terms of existing water quality or soil quality standards of China or any organisation or country. Please use tables or clustered bar graph on how these results compare with existing quality standards and what are the implications.
Furthermore, the significance and meaning of the results need to be expressed in terms of what other similar studies have found. Are there similarities or dissimilarities and what are the implications for sustainable management of affected catchments? What do these results mean for water quality management in the study area or in China?
The conclusion does not give a good synthesis and integration of the work. Please link each conclusion to specific research objectives, and specify implications. After specifying the conclusions, can you pls indicate fruitful research areas for future studies
The Abstract needs slight reformulation and must also be submitted for language editing. In addition, can you briefly specify some of the main recommendations in this abstract

Author Response
Responses to the Reviewers Comments
#To Reviewer 3:
We sincerely appreciate for your patient reading and constructive comments on this MS. According to your valuable comments and suggestions, the MS was revised very carefully to improve the quality and readability. Please see a detailed, point-by-point response:
The language used is of low quality therefore after making revisions, please submit the whole manuscript to a language editor for refinements and editorials.
Response 1: Thanks for your advice. The whole manuscript has been edited by ACS Authoring Service (25B8-117D-D44B-97FB-1432).
The rationale and justification for this study is not clearly specified, is this a pilot study because you collected only four samples to represent the whole basin or catchments. I suggest that you give a justification of this limited sampling. Maybe its a pilot study and will be followed by a bigger study, however that should be clearly identified and justified upfront.
Response 2: I'm sorry for this confusion, and this is not a pilot study for the whole basin. The objective of the current study was to the four basins which were long-term polluted by RDW discharge, not a study of the whole. Four small basins (B1-B4) was selected because long-term polluted by RDW discharge. The area of the selected small basins (B1-B4) were all limited (223 - 781 m2 Table 1), and “Each sediment sample (S1-S4) was collected from 5 random spots along the shore of the selected basin with a sampling grab at depth of 0-10 cm and then mixed homogenously (Pg 2, Ln 39 - 41)” to ensure the representativeness of sediment sample.
The presentation of the results needs improvement. The concentration of pollutants expressed in the results must also be interpreted in terms of existing water quality or soil quality standards of China or any organisation or country. Please use tables or clustered bar graph on how these results compare with existing quality standards and what are the implications.
Response 3: Thanks for your kind advice. The comparison between ammonium concentration in the results and the Chinese environmental quality standards for surface water has been added in the Figure 2 and Figure 4. Correspondingly, the related description and discussion were added in manuscript as follows:
“ is a useful index for an internal ammonium release risk comparison which during the S1 incubation increased with the increase in the sediment-to-water ratio, i.e., 1.42, 1.99, 2.07 and 3.36 mg·L-1, at sediment-to-water ratios of 1:20, 1:10, 1:5 and 1:1, respectively, all of which breached the class â…¢ surface water-quality standard in China (Figure 2).” (Pg 6, Ln 17 - 18)
“And all of the maximum breached the class â…¢ surface water-quality standard in China (Figure 4).” (Pg 8, Ln 34 - 35)
The significance and meaning of the results need to be expressed in terms of what other similar studies have found. Are there similarities or dissimilarities and what are the implications for sustainable management of affected catchments? What do these results mean for water quality management in the study area or in China?
Response 4: The literature on internal ammonium release from sediments RDW as a result of long-term RDW discharge is rare, and we haven’t found effective data for comparison. Some water quality management suggestions were supplied in conclusion “dredging was not suggested for RDW-polluted sediments except in response to emergency. The excessive ammonium in the selected catchment was mainly a result of discharge of untreated and centralized black water in RDW. The centralized black waters in rural community were highlighted for separate treatment or reuse to maintain aqueous ammonium at a safe level” (Pg 12, Ln 15 - 19)
The conclusion does not give a good synthesis and integration of the work. Please link each conclusion to specific research objectives, and specify implications. After specifying the conclusions, can you pls indicate fruitful research areas for future studies.
Response 5: Thanks for your kind suggestion. The conclusion was revised as “The risk of ammonium release from sediment during the dry season was found to be significantly greater than that during the wet season in this study. Internal ammonium release from sediment was a slow and long process and the risk of ammonium release from sediment could be overestimated by the theoretical calculation using tested exchangeable nitrogen. Sediments can serve as “source” or “sink” regulating the ammonium balance in an aquatic ecosystem cycle, such as the “sink→source” effect of a high concentration of RDW discharge. Though all of the selected sediments were heavily polluted as a result of long-term RDW discharge, their relative contribution of internal loading was generally less than that of external pollution. Hence, dredging was not suggested for RDW-polluted sediments except in response to emergency. The excessive ammonium in the selected catchment was mainly a result of discharge of untreated and centralized black water in RDW. The centralized black waters in rural community were highlighted for separate treatment or reused to maintain aqueous ammonium at a safe level. We highly call for more studies on nitrogen circle contributed by the RDW discharge and proper treatment technology for centralized black waters.” (Pg 12, Ln 8 - 20)
The Abstract needs slight reformulation and must also be submitted for language editing. In addition, can you briefly specify some of the main recommendations in this abstract.
Response 6: Thanks for your suggestion. The whole manuscript has been edited by ACS Authoring Service (25B8-117D-D44B-97FB-1432). And the Abstract was reformulated as “Given long-term decentralized and centralized rural domestic wastewater (RDW) discharge, nitrogen is continuously depositing in sediments. RDW discharge is assumed to be an important source of ammonium in surface water; however, the effect of long-term RDW discharge on nitrogen pollution in sediments remains unknown. Batch incubations were conducted to investigate the characteristics of internal ammonium loading from long-term polluted sediments by RDW discharge. Four sediments were demonstrated to be heavily polluted by long-term RDW discharge, with total nitrogen (TN) values of 5350, 8080, 2730 and 2000 mg·kg-1, respectively. The internal ammonium release from sediment was a slow and long process, and the risk of ammonium release from sediment during the dry season was significantly greater than that during the wet season. Though all selected sediments were heavily polluted by long-term RDW discharge, the relative contribution of internal ammonium loading from sediments was generally lower than that of external pollution. Hence, dredging is not suggested for RDW-polluted sediments except in response to an emergency. The excessive ammonium in the selected catchment was mainly from untreated and centralized black water in RDW. Centralized black waters in rural community are highlighted to be separately treated or reused to maintain ammonium content at a safe level” (Pg 1, Ln 20 - 34).

Round 2
Reviewer 2 Report
Issues previously raised have been addressed. No further comments.